# Detailed Speciation of Non-Methane Volatile Organic Compounds in Exhaust Emissions from Diesel and Gasoline Euro 5 Vehicles Using Online and Offline Measurements

**DOI:** 10.3390/toxics10040184

**Published:** 2022-04-08

**Authors:** Baptiste Marques, Evangelia Kostenidou, Alvaro Martinez Valiente, Boris Vansevenant, Thibaud Sarica, Ludovic Fine, Brice Temime-Roussel, Patrick Tassel, Pascal Perret, Yao Liu, Karine Sartelet, Corinne Ferronato, Barbara D’Anna

**Affiliations:** 1Aix Marseille Univ, CNRS, LCE, UMR 7376, 13331 Marseille, France; vkostenidou@gmail.com (E.K.); brice.temime-roussel@univ-amu.fr (B.T.-R.); 2French Agency for Ecological Transition, ADEME, 49000 Angers, France; boris.vansevenant@univ-eiffel.fr; 3Univ Lyon, Université Claude Bernard Lyon 1, CNRS, IRCELYON, 69626 Villeurbanne, France; alvaro.mval10@gmail.com (A.M.V.); ludovic.fine@ircelyon.univ-lyon1.fr (L.F.); corinne.ferronato@ircelyon.univ-lyon1.fr (C.F.); 4Univ Gustave Eiffel, Univ Lyon, AME-EASE, 69675 Lyon, France; patrick.tassel@ifsttar.fr (P.T.); pascal.perret@ifsttar.fr (P.P.); yao.liu@ifsttar.fr (Y.L.); 5CEREA, Ecole des Ponts ParisTech, EdF R&D, 77455 Marne-la Vallée, France; thibaud.sarica@enpc.fr (T.S.); karine.sartelet@enpc.fr (K.S.)

**Keywords:** Euro 5, emissions, PMF, NMVOCs, diesel, gasoline, PTR-ToF-MS, ATD-GC-MS, oxygenated compounds, BTEX, alkanes, alkenes, IVOCs

## Abstract

The characterization of vehicle exhaust emissions of volatile organic compounds (VOCs) is essential to estimate their impact on the formation of secondary organic aerosol (SOA) and, more generally, air quality. This paper revises and updates non-methane volatile organic compounds (NMVOCs) tailpipe emissions of three Euro 5 vehicles during Artemis cold urban (CU) and motorway (MW) cycles. Positive matrix factorization (PMF) analysis is carried out for the first time on proton transfer reaction time-of-flight mass spectrometer (PTR-ToF-MS) datasets of vehicular emission. Statistical analysis helped to associate the emitted VOCs to specific driving conditions, such as the start of the vehicles, the activation of the catalysts, or to specific engine combustion regimes. Merged PTR-ToF-MS and automated thermal desorption gas chromatography mass spectrometer (ATD-GC-MS) datasets provided an exhaustive description of the NMVOC emission factors (EFs) of the vehicles, thus helping to identify and quantify up to 147 individual compounds. In general, emissions during the CU cycle exceed those during the MW cycle. The gasoline direct injection (GDI) vehicle exhibits the highest EF during both CU and MW cycles (252 and 15 mg/km), followed by the port-fuel injection (PFI) vehicle (24 and 0.4 mg/km), and finally the diesel vehicle (15 and 3 mg/km). For all vehicles, emissions are dominated by unburnt fuel and incomplete combustion products. Diesel emissions are mostly represented by oxygenated compounds (65%) and aliphatic hydrocarbons (23%) up to C_22_, while GDI and PFI exhaust emissions are composed of monoaromatics (68%) and alkanes (15%). Intermediate volatility organic compounds (IVOCs) range from 2.7 to 13% of the emissions, comprising essentially linear alkanes for the diesel vehicle, while naphthalene accounts up to 42% of the IVOC fraction for the gasoline vehicles. This work demonstrates that PMF analysis of PTR-ToF-MS datasets and GC-MS analysis of vehicular emissions provide a revised and deep characterization of vehicular emissions to enrich current emission inventories.

## 1. Introduction

Ambient air pollution is a complex mixture of gaseous and particulate pollutants and represents the fifth major cause of disease and death in the world with an estimated 4.2 million premature deaths and 103.1 million lost years of healthy life in 2015 [1]. Air pollutants arise from a variety of sources, both biogenic and anthropogenic. Among them, vehicle emissions represent about 52% of total nitrogen oxides (NO_x_), 38% of CO_2_, and 40% of black carbon (BC) in France during 2019 [2]. Moreover, the source apportionment of VOCs in Paris identified vehicle exhaust emissions as a large VOC source, representing 15% of the total VOC mass [3]. Since 1992, to lower the impact of vehicle exhaust emissions on air quality, European instances have established standards labeled as Euro 1–6, introducing emission limits to critical pollutants such as carbon monoxide (CO), total hydrocarbon content (THC), non-methane hydrocarbons (NMHC), NO_x_, THC + NO_x_, particulate matter (PM), and particle number (PN). Over the years, stricter limitations have forced manufacturers to develop better formulations of fuel blends, more efficient engines, and better aftertreatment systems for both gasoline and diesel vehicles, leading to the implementation of various technologies such as diesel oxidation catalysts (DOCs) [4], three-way catalysts (TWCs) [5], and diesel particulate filters (DPFs) [6]. The Euro 5 standard, more particularly, generalized the DPF for diesel vehicles. Those improvements led to a decrease of 67% in NO_x_ emissions and 72% of BC emissions from road transport between 1990 and 2019 [2] in France. Moreover, the most recent diesel vehicles equipped with DPF emit less primary particles than their gasoline homologues [7,8,9].

In addition to these primary emissions, vehicle exhausts contribute to anthropogenic SOA (ASOA), which may become preponderant in urban areas [10] but also at the global scale [11]. Recent studies on the oxidative potential of PM have shown a link between OA chemical composition and its potential health impact [12,13,14,15], highlighting the importance of understanding and controlling specific PM precursors, particularly the anthropogenic ones. Different approaches exist to understand SOA formation from diesel and gasoline vehicles, either based on bottom-up studies, using unburnt fuels or dilute vehicle exhaust emissions as emission surrogates in chamber experiments, or based on top-down studies, focusing on the chemical composition of ambient OA coupled with source apportionment techniques [16]. Recent advances in this field suggest the importance of a detailed speciation of SOA precursors, since some categories of compounds such as IVOCs, despite their low fraction of the total VOC emissions (1–4% of the total NMHC emissions for gasoline vehicles [17,18,19] and 1.5% of the total NMHC emissions for diesel vehicles [20]), contribute at least as much to SOA formation than traditional precursors, i.e., single-ring aromatics [10,21]. 

Nonetheless, a complete view on the VOC composition from diesel and gasoline vehicles is a challenging task. Many factors, such as fuel composition (e.g., ethanol fraction [22,23]), injection technology (PFI or GDI [24]), engine capacity, aftertreatments systems [25,26], cold temperature [27,28], driving conditions (cold start [17] or load [29]), and mileage [30], can drastically influence the emissions, both quantitatively and qualitatively. Moreover, emissions due to rapid transient phenomena associated with vehicle driving patterns (starts and restarts [31], tip-in [32]), catalyst light-off [33,34], or particulate filter regenerations [35], which occur over time periods ranging from a few seconds to a few hundred seconds, are difficult to evaluate. Traditionally, the extensive characterization of VOCs and IVOCs is carried out by offline measurements (e.g., gas chromatography [19,20,36,37] or two-dimensional gas chromatography [17,38]), and lacks of time-resolved information on the emissions. Online measurements, on the other hand, are often based on chemical ionization mass spectrometry (CI-MS), focusing on compounds of interest, such as nitrogen-containing compounds [39,40], carbonyls, benzene, toluene, ethylbenzene, and xylenes (BTEX) [41]. However, VOC identification and quantification by CI-MS techniques such as PTR-MS [42] are hindered by fragmentation reactions which occur in the drift tube, and by their limited mass resolution, which prevents the separation of isobaric signals. Erickson et al. [43] and Gueneron et al. [44] studied the fragmentation patterns of hydrocarbons families detected in diesel and gasoline exhaust emissions using a PTR-MS at two reduced electric field conditions of 80 and 120 Townsend (Td). Fragmentation was drastically reduced at 80 Td compared to 120 Td, and most of the alkenes and aromatic compounds did not fragment at all or mostly yielded their molecular ions. On the contrary, alkanes, cycloalkanes, and bicycloalkanes underwent extensive fragmentation even at 80 Td, complicating the differentiation between small alkenes and alkane fragments. Erickson et al. [43] proposed a new method to measure IVOCs using a thermal desorption sampler integrated into a PTR-MS, providing quantitative information on the total abundance of long-chain alkanes and aromatics species in diesel exhausts. Although this method allows for the quantification of IVOCs, its time resolution (approximately 15 min) does not give information on the temporal variation of IVOC emission. Other studies using instruments with a higher mass resolution could separate isobaric signals. Pieber et al. [45] measured the general gas phase composition of GDI vehicles using a PTR-ToF-MS (PTR-TOF-8000, Ionicon Analytik GnbH, Innsbruck, Austria; [46,47]) operated at 140 Td. They could measure 65% of the total NMVOC signal, including oxygenated species, such as carbonyls or acids, as well as nitrogen-containing compounds. However, the intensive fragmentation of alkanes, alkenes, cycloalkanes, and substituted monoaromatics is expected at such a high reduced electric field (E/N) [44], complicating their identification. The potential of the high time resolution of the PTR-ToF-MS was not highlighted in this study. More generally, time-resolved studies on PTR-ToF-MS datasets of vehicle exhaust emissions are lacking. Yet, highly time-resolved measurements have recently shown their usefulness, not only in VOCs source apportionment in urban and rural aeras [48,49,50,51], but also for factor analysis of SOA formation [52]. They are generally conjugated with receptor models, particularly PMF [53]. To our knowledge, the present study is the first applying PMF analysis to PTR-ToF-MS datasets of gasoline and diesel vehicle exhaust emissions.

The present work describes the results of the first PMF analysis applied to the highly time-resolved PTR-ToF-MS measurements (1s resolution) of three Euro 5 vehicles sampled on a roll-bench chassis dynamometer during the Artemis driving cycles. This analysis aims to untangle the multiple factors characterizing modern vehicle emissions. Moreover, the paper aims to provide an exhaustive inventory of the NMVOC EFs of one diesel, one PFI, and one GDI vehicle, achieved by merging datasets from online PTR-ToF-MS measurements with complementary offline GC-MS analysis. Monitored compounds include saturated and unsaturated hydrocarbons, as well as oxygen- and nitrogen-containing compounds. The combined techniques help to span a large range of compounds, starting from C_1_-oxygenated compounds to IVOC pollutants. Up to 147 compounds have been identified and quantified.

## 2. Materials and Methods

### 2.1. Vehicle Characteristics

Three Euro 5 passenger vehicles were tested: a diesel vehicle equipped with an oxidation catalyst and a fuel-borne catalyst diesel particle filter (FBC-DPF), as well as a gasoline port fuel injection (PFI) vehicle and a GDI vehicle, both equipped with TWCs. Their detailed characteristics are described in Table 1.

The diesel and the PFI vehicles were tested during a field campaign in 2018, while the GDI vehicle was tested in another field campaign in 2019. More details about the test procedure for each vehicle are given in Appendix A. The passenger vehicles were either rented from a local car rental company or privately owned, and their mileage ranged from 27,712 to 103,000 km. All three vehicles were fueled with commercial diesel and gasoline SP95-E10 purchased from the same gas station to minimize variability of the fuel composition during the tests. Diesel and gasoline fuel headspaces were analyzed by PTR-ToF-MS and their compositions are presented in Appendix A, respectively. All three vehicles were tested using the Artemis European driving cycle [54]. This choice was motivated by the need for us and also for modelers to separate emissions as a function of the driving conditions (cycle speed) and the geography (urban, rural, or motorway). Therefore, experiments focused on the CU and the MW cycles to account for the engine start during both cold and hot conditions and to measure the efficiency of the depollution technologies on a broad range of driving conditions. 

### 2.2. Experimental Setup

A schematic of the experimental setup is described in Figure 1. Experiments were carried out at the Environment, Planning, Safety, and Eco-design Laboratory (EASE) of the Gustave Eiffel University. Vehicles were tested on a chassis dynamometer test bench. Road loads of the dynamometer are described in Table 1. The total exhaust flow was sampled simultaneously using two dilution systems. The first method used filtered ambient air through a constant volume sampler (CVS) set at a total flowrate of 9–11 m^3^/min for the CU and MW cycles, respectively, and was dedicated to the analysis of regulated pollutants, such as THC, NO_x_, CO, and also CO_2_. All gas-phase-regulated compounds were monitored in parallel by online analysis and bag collection. More details on the chassis dynamometer configuration and the CVS are given elsewhere [55,56,57].

Secondly, a fraction of the exhaust flow was sampled through a 5–6 m-long stainless-steel line with a 10 mm inner diameter heated at 120 °C using a Sapelem ejector (with hot air) of one stage, allowing us to sample a constant volume of exhaust gases. During this step, exhaust gases were diluted with dehumidified and filtered air (using HEPA filters and activated carbon). Total dilution ratios selected for the different vehicles and instruments are presented in Table 1. Online and offline measurements of both the gas- and particle-phase emissions were carried through this second sampling line using a suite of instrumentation. VOC and IVOC measurements were conducted using a PTR-ToF-MS completed by ATD-GC-MS offline measurements. The instrumentation used to analyze the particle phase is described elsewhere [9].

### 2.3. Measurement Techniques for Gaseous Pollutants

#### 2.3.1. PTR-ToF-MS

VOC online measurements were carried out using a PTR-ToF-MS 8000 (Ionicon Analytik, Austria) [46,47] in H_3_O^+^ mode with a time resolution of 1 s. Exhaust gases were sampled through a two-meter-long silcosteel line with a 1 mm inner diameter heated at 120 °C with a flowrate of 400 cm^3^/min. The silcosteel line was directly connected to the stainless steel heated line. Two additional dilution steps were applied to the sampling line before the PTR-ToF-MS inlet: the first one using clean air generated with a Sonimix zero air generator SX-3057, and the second one using dry nitrogen from the fast GC system of the instrument. The total dilution ratio for each vehicle is described in Table 1, while dilutions at each step are described in Appendix A. These two dilution steps were useful to avoid saturation of the signal of the PTR-ToF-MS and minimize the relative humidity of the samples. The drift tube was kept under controlled conditions of pressure, temperature, and voltage (2.04 mbar, 383 K, and 395 V for the diesel and gasoline PFI vehicles, and 2.26 mbar, 393 K, and 395 V for the gasoline GDI vehicle), resulting in E/N of 116 and 108 Td, respectively.

Raw PTR-ToF-MS data were post-processed using the data analysis package “Tofware” (version 2.5.10, [58]), running in the Igor Pro (Wavemetrics, OR, USA) environment. The molecular formula assignment could be carried out thanks to the high mass resolution of the PTR-ToF-MS (m/∆m ≈ 3500). Tentative ion assignment was based on ATD-GC-MS offline measurements and on literature reports on vehicular emissions [36,37,43,44,59]. Identified ions were classified in 13 ion families, including alkanes/alkenes, cycloalkanes, bicycloalkanes, monoaromatics, naphthenic monoaromatics, dihydronaphthalenes, naphthalenes, alcohols, carbonyls, unsaturated carbonyls, acids, other oxygenated compounds, and nitrogen-containing compounds.

The post-process step resulted in a matrix containing the time series in count per second (cps) of each ion identified. Data in cps were then corrected for the background, which was measured before the beginning of each cycle. Background corrected data were finally converted into ppb using the transmission function of our PTR-ToF-MS. The VOC proton reaction rate constants with H_3_O^+^ were either directly obtained or interpolated by linear regression using data from A. Wisthaler (personal communication) based on various proton reaction rate constants [60,61]. A proton reaction rate constant of 2 × 10^−9^ cm^3^/s was used for VOCs with no available data.

#### 2.3.2. ATD-GC-MS

Off-line VOC and IVOC measurements were collected from the heated line by sampling diluted exhaust gases through stainless steel tubes filled with Tenax TA at a flow rate of 45 cm^3^/min. The samples were collected during the entire driving cycle and were further analyzed by automatic thermal desorption (Markes Unity Thermodesorber) coupled with a GC6890 gas chromatograph from Agilent fitted with the MS5973 mass spectrometer from Agilent (ATD-GC-MS). The thermal desorption system consists of a two-stage desorption. During the first desorption step, the compounds were desorbed by heating the stainless steel tubes at 300 °C under a helium flowrate of 35 cm^3^/min and were then condensed on a trap filled with adsorbent and maintained at 15 °C. During the second desorption, the second trap was flash-heated to 305 °C with an outlet split of 15 cm^3^/min for a rapid introduction of the compounds into the chromatographic column. The chromatographic column was an Agilent HP1MS (30 m, 0.25 mm, and 0.25 μm) used in thermal gradient mode from 40 °C to 320 °C. The mass spectrometer operated in the scanning mode at an electron ionization of 70 eV. Mass spectral data were acquired over a mass range of 33–350 amu. The qualitative identification of compounds was based on the match of the retention time and confirmed by matching their mass spectra with those of standards and from the NIST mass spectral library.

Quantification was conducted by the external standard method using different certified commercial mixtures from Sigma-Aldrich containing linear and branched alkanes, cyclo- and bicycloalkanes, and alkyl monoaromatics. Known amounts (1 μL) of standard solutions of VOCs and IVOCs were introduced into cleaned Tenax TA tubes using an automatic heated GC injector. The calibration tubes were analyzed under the same conditions, as previously mentioned.

The chromatograms obtained from the exhaust analysis showed an unresolved complex mixture, mainly composed of coeluted hydrocarbons which cannot be further separated by single-dimensional GC. Thus, the alkanes (linear and branched) were quantified by a SIR-based response factor of these compounds using the fragment *m*/*z* 57. The fragments *m*/*z* 84 and *m*/*z* 83 were used for the quantification of cyclohexane and for the other cycloalkanes, respectively. The fragment *m*/*z* 78 was used for the quantification of benzene and *m*/*z* 91 for toluene, ethylbenzene, and xylenes. The fragments *m*/*z* 105, 119, and 134 were used for the quantification of alkylaromatics. The fragment *m*/*z* 128 was used for naphthalene.

### 2.4. PMF Analysis

PMF is an unmixing bilinear model used to investigate the source contributions and the temporal evolutions of environmental datasets [53]. Here, the PMF was performed on the PTR-ToF-MS data matrix from each type of vehicles and driving cycles separately. All the repetitions of a cycle were concatenated in a unique matrix to increase the number of samples used. The error matrix was calculated using Equation (1) ([62,63]) with $\Delta \left(I_{CC=off}-I_{CC=on}\right)$ corresponding to the error on the background-corrected signal in cps, $I_{CC=on}$ corresponding to the background signal in cps, $I_{CC=off}$ corresponding to the sample signal in cps, $\tau_{CC=on}$ corresponding to the dwell time during the background measurement, and $\tau_{cc=off}$ corresponding to the dwell time during the sample measurement:(1)$$\Delta \left(I_{CC=off}-I_{CC=on}\right)=\sqrt{\frac{I_{CC=on}}{\tau_{CC=on}}+\frac{I_{CC=off}}{\tau_{cc=off}}}$$

Data matrices were then filtered by rejecting each ion with an average signal-to-noise ratio lower than two. The PMF algorithm was solved with the multilinear engine 2 (ME-2; [64]) using the software SoFi (Version 6.8, [65]) running in the Igor Pro (Wavemetrics, OR, USA) environment. PMF runs were carried in the robust mode with a number of factors ranging from 1 to 10. The downweighting step was skipped as the dataset was previously filtered from signals with an averaged signal-to-noise (S/N) ratio lower than two. Most representative solutions were finally chosen based on theoretical and physical considerations, as detailed in the Appendix A.

### 2.5. Calculation of EFs 

VOC emission factors (EFs) were calculated using both the PTR-ToF-MS and GC-MS datasets for the 3 tested vehicles. Neither PTR-ToF-MS nor GC-MS can measure alkanes < C_5_. PTR-ToF-MS can detect alkene signals, but ethylene measurement is not quantitative as the C_2_H_4_^+^ ion signal comes from the charge transfer between ethylene and residual O_2_^+^ [66]. Thus, EFs were only calculated for alkanes > C_6_ and alkenes > C_3_. Furthermore, methane and ethylene, whose summed EFs can account for 10 to 20 mg/km [67], were not measured by our techniques. Moreover, the analysis of branched alkanes by ATD-GC-MS was only carried in 2019 for the GDI vehicle. 

EFs in mg/km for both PTR-ToF-MS and GC-MS data were calculated using Equation (2):(2)$$EF=C_{x,average}\times \frac{T_{cycle}\times Q_{exhaust,average}\times DR}{D}$$
where $T_{cycle}$ is the cycle duration in seconds (993 s for the UC and 1067 s for the MW cycle); $C_{x,average}$ is the averaged mass concentration of pollutant x in mg/m^3^; $Q_{exhaust,average}$ is the averaged exhaust flow rate at the tailpipe in m^3^/s; DR is the total dilution ratio before the entrance of the instrumentation, as described in Table 1; and D is the distance travelled during the cycle in km (4.874 km for the Artemis UC and 29.547 km for the Artemis MW cycle).

The EF uncertainty is mainly due to the error on $C_{x,average}$. Relative uncertainty on the concentration measured by the PTR-ToF-MS is approximately 25%, and is considered as the square root of the sum of squared uncertainties on the transmission and the proton reaction rate constants [60,68]. On the other hand, relative uncertainty on the concentration measured by the ATD-GC-MS is approximately 20% [69]. 

As no Tenax cartridges were sampled during the diesel vehicle CU cycle, GC-MS data for alkanes were not available for this cycle. Moreover, it is usually not possible with a PTR-TOF-MS to distinguish alkane fragments from alkene as they both produce (C_x_H_y_)H^+^ ions. In this work, the part of the signal attributed to alkenes was discriminated based on the PMF results. The total alkane signal in cps was then calculated based on the remaining part of the (C_x_H_y_)H^+^ ion signals. The alkane profile was then reconstructed up to C16 (heaviest alkane signal detected by the PTR-ToF-MS during the CU cycle) using the GC-MS alkane profile measured during the MW cycle. Alkanes > C_16_ were not included as their emissions were not repeatable.

## 3. Results and Discussion

### 3.1. PMF Analysis

Figure 2 presents the solution factors derived by PMF analysis for (a) the diesel car during the CU cycle, (b) during the MW cycle, and (c) the GDI car during the CU cycle. PMF input matrices consisted of four, two, and six concatenated cycle repetitions, respectively, for the diesel CU cycle, the diesel MW cycle, and the GDI CU cycle. PMF analysis for the PFI CU cycle was limited by the absence of repetition for this cycle, and the saturation of signals of interest, such as BTEX and alkenes at *m*/*z* 45, 57, 93, 107, and 121. Still, these results are presented in Appendix A. Moreover, the low concentrations measured during the MW cycle for the PFI and GDI vehicles resulted in low S/N ratios that prevented statistical analysis for these cycles. 

#### 3.1.1. Diesel Vehicle

Diesel cold urban cycle

The PMF analysis of the diesel car emissions during the CU cycle resulted in four distinct factors. Their temporal variation is presented on the left side of Figure 2a., while factors to species contribution ratios are presented on the right side of Figure 2a. These factors to species contribution ratios are meant to illustrate which factor mostly influences the temporal variations of each species, depending on its carbon, oxygen, and nitrogen number. The factors to species contribution ratios summarized in Figure 2a are listed in Appendix A for the factors 1, 2, 3, and 4, respectively.

Factor 1 temporal variation is characterized by the highest VOC concentrations emitted essentially during the first 400 seconds of the cycle. The main emission peaks reach 20–30 ppm and are well correlated with the accelerations of the vehicle. These pollutants are emitted before the activation of the DOC, as their emissions coincide with CO measured inside the CVS, as presented in Appendix A. The major contributors to this factor are C_x_H_y_ and C_x_H_y_O species with carbon numbers < C_5_. They are characteristic of unburnt fuel and incomplete combustion products. Long-chain alkanes, cycloalkanes, and bicycloalkanes, i.e., major components of diesel fuel [18], are absent of this factor, although they were measured in the fuel headspace, as presented in Appendix A. This behavior is attributed to condensation losses of lower volatility species on the cold engine manifold, aftertreatment systems, and exhaust line [29,35]. 

Factor 2 exhibits persistent emission of VOCs along the whole cycle with an average concentration of 2 ppm, and emission peaks associated with the main accelerations around 5 ppm. A slight decrease in the concentrations is observed during the “free-flow urban” and the “flowing, stable” phases [54] between 300 to 500 seconds and from 900 seconds to the end, respectively. This behavior suggests that the driving conditions have an impact on these emissions. Compounds such as ethylene, benzene, acetaldehyde, and acrolein, which are found in factor 1, also contribute to factor 2. Their presence highlights that the DOC efficiency varies depending on compounds type and their concentration [4]. Nitromethane and formic acid are mostly present in factor 2 (with factors to species contribution ratios of 75 and 53%, respectively), suggesting that aftertreatment systems have little or no impact on them during the CU driving cycle. Nitromethane emissions are indeed known to be associated with engine operation conditions and do not depend on DOC activity [39].

Factor 4 is measured only during two of the four CU cycles. It appears after 400 s from the start of the cycle with an emission peak around 10 ppm and continues until the end of the cycle. C_x_H_y_O_2–4_ species, such as acetic acid and maleic anhydride, highly contribute to this factor at 99 and 100%, respectively. These compounds are possible by-products of incomplete oxidation by the DOC [33,34] and the DPF [70]. Thus, this factor underlines the impact of the aftertreatment systems on the emission of oxidized species. Factor 4 also includes C_x_H_y_ ion fragments, with contribution ratios which increase alongside the carbon number. These compounds could be associated with the desorption of heavier hydrocarbons from the cold engine manifold and aftertreatment systems, possibly from lubricant oil droplets occasionally emitted at the start of the vehicle, and their partial treatment by the DOC and the DPF. These droplets are mostly composed of long-chain cycloalkanes [71] and their emission is not a repeatable phenomenon with no clear correlation with engine load, cycle speed, or acceleration, as reported for the same vehicles by Kostenidou et al. [9].

Factor 3 is characterized by a spread-out peak reaching 10 ppm just before the 400th second of the cycle. Most of the emissions associated with this factor occur during the “free-flow urban” section of the Artemis CU [54] and, to a lesser extent, at the end of the cycle during the “flowing, stable” section. C_x_H_y_ species such as alkanes, cycloalkanes, bicycloalkanes, monoaromatics, and naphthenic monoaromatics contribute to this factor. These compounds are major components of diesel fuel [18,43] and are measured in the fuel headspace presented in Appendix A, but they are not emitted with other unburnt compounds at the beginning of the cycle (factor 1). They are supposedly desorbed during the warming-up of the engine manifold and aftertreatment systems [29,35]. This factor is associated with C_x_H_y_O species, such as saturated, unsaturated, and aromatic carbonyls, with 3 to 8 carbon atoms. These oxidized compounds could be by-products of incomplete oxidation by the DOC during its light-off [33,34].

Diesel motorway cycle

PMF analysis of the diesel car emissions during the MW cycle resulted in five distinct factors. As for the CU analysis, temporal variations of the five factors for the MW cycle are presented on the left of Figure 2b, while factors to species contribution ratios are presented on the right of Figure 2b. The factors to species contribution ratios summarized in Figure 2b are also listed in Appendix A for factors 1, 2, 3, 4, and 5, respectively.

Factor 1 shares some features with CU’s factor 1, as it occurs only at the beginning of the cycle, during the first 100 seconds of the MW cycle, with concentration peaks reaching 8 ppm. However, its composition appears to be a mixture of compounds found in CU’s factors 1 and 2. While these compounds are not efficiently converted during the CU cycle, they are fully removed during the MW cycle after the first 200 seconds. Factor 1 is observed during the first MW cycle, suggesting that the aftertreatment systems are not fully operational at this time. Nitromethane emissions are still present at the start of the MW cycle, but they are not observed anymore at speeds above 100 km/h.

Factors 2 and 3 present similarities with respect to cold start’s factor 3 temporal variation and composition. They are emitted one after another at the beginning of the cycle, with peaks at 10 and 2 ppm occurring at 150 and 200 seconds, respectively. Both factors contain C_x_H_y_ species such as alkanes, cycloalkanes, and bicycloalkanes fragments. Incomplete combustion products such as C_3_ to C_5_ carbonyls are strongly correlated to MW’s factor 2, while MW’s factor 3 contributes to monoaromatics and oxidized species, such as maleic anhydride, a potential by-product of incomplete oxidation by the DOC [72].

Factor 4 is a small factor (reaching 1 ppm) which generally correlates to the cycle speed and appears at speeds higher than 80 km/h. This factor is highly repeatable, and it is strongly associated with oxidized species, such as unsaturated carbonyls, phthalic anhydride, benzoquinone, maleic anhydride, and, to a lesser extent, some alkane fragments and monoaromatic compounds. The latter represent the fraction of the emissions that is not converted by the aftertreatment systems, while oxidized species are potential by-products of incomplete oxidation by the DOC.

Finally, factor 5 exhibits relatively high emissions around 2 ppm occurring during the last section of the MW cycle when the speed reaches 140 km/h, and also at the beginning of the second MW cycle where it increases up to 6 ppm. Similar to CU’s factor 4, MW’s factor 5 is mainly associated with acetic acid and its emission is not repeatable. Following our precedent hypothesis, it could be linked to particular operations of the aftertreatment systems. 

#### 3.1.2. GDI Vehicle

Figure 2c shows the averaged temporal variations and relative contributions of the six PMF factors calculated during the CU cycle for the Euro 5 GDI vehicle. Gasoline NMVOC emissions mostly occur at the beginning of the cycle, to such a degree that GDI cold start emission control has become a major issue in recent years [26,73]. All the six factors exhibit an emission peak during the first acceleration. This behavior can also be observed for the PFI vehicle, as presented in Appendix A. Factor 2 shows good correlation with vehicle acceleration, while the other factors show similar evolution as the diesel vehicle’s factors and seem to be correlated with the temperature rise in the engine and aftertreatment systems. Deconvolution of the emitted compounds in different factors seems to be closely related to the pollutants carbon and oxygen numbers. The factors to species contribution ratios, summarized in Figure 2c, are listed in Appendix A for factors 1, 2, 3, 4, 5, and 6, respectively.

Factor 1 is characterized by a unique intense peak synchronized with the first acceleration reaching approximately 150 ppm. Small unsaturated hydrocarbons, such as alkenes from C_2_ to C_6_; dienes from C_3_ to C_6_; and signals from (C_4_H_4_)H^+^, (C_5_H_6_)H^+^, and (C_6_H_8_)H^+^ ions (which could also correspond to highly unsaturated hydrocarbons, such as alkynes and cycloalkadienes), are associated with this factor. These compounds are similar to those found in the most volatile fraction of the fuel [37,59], and are also measured in the headspace of the fuel (Appendix A); therefore, they have been associated with unburnt fuel.

Factor 2 is mainly emitted during strong accelerations and disappears after the first 600 seconds of the cycle. Benzene, alkane, and cycloalkane fragments are associated with this factor. The predominance of benzene among other monoaromatics suggest that factor 2 corresponds to fuel-rich combustion. Indeed, it has been shown that, in such regime, dealkylation of alkylbenzenes takes place, leading to the formation of benzene, toluene, and oxidized by-products [74,75]. Concomitant CO emissions during the same strong accelerations (Appendix A), typically associated with fuel-rich combustion, corroborate this assumption. The oxygen deficiency also leads to lower conversion of hydrocarbon species [5], explaining the presence of alkane and cycloalkane fragments.

Factor 3 presents a very similar pattern to factor 1 but exhibits a broader emission peak occurring a few seconds later. Ethanol is associated with this factor, followed by several cycloalkanes from C_6_ to C_9_, aldehydes from C_1_ to C_5_, and unsaturated aldehydes (such as acrolein). These oxidized compounds are associated with incomplete combustion products.

Factors 4 and 5 both present a first peak at 37 and 45 seconds, respectively, followed by a second spread-out peak between 50 and 300 seconds. Although the two factors have similar temporal variations, their chemical composition is quite different. Factor 4 is associated with C_7_–C_9_ aromatic compounds and factor 5 to C_9_–C_11_ aromatic compounds. The same trend is observed for the dihydronaphthalenes and naphthenic monoaromatic compounds, for which the contribution to factor 4 decreases with increasing carbon number, to factor 5’s benefit. Oxidized compounds, such as oxalic acid, benzaldehyde, acetophenone, and other oxygenated compounds, are also associated with factor 4 and 5. As for the diesel vehicle, this behavior is supposedly due to the progressive desorption of lower volatility unburnt fuel components and particular operations of the TWC during engine and aftertreatment systems warm-up.

Finally, factor 6 is characterized by relatively low and constant emissions, with a peak at 2 ppm occurring 20 seconds after the first acceleration. This factor is associated with lower volatility compounds, such as C10 and C11 aromatics, cycloalkanes, bicycloalkanes, indane, indene, styrene, tetraline, dihydronaphthalene, and naphthalene, but also with oxidized species, such as oxalic acid, acetophenone, and acrolein. Although they represent a small fraction of the total emitted species during the whole cycle, their contribution becomes preponderant during the 400 last seconds, after the TWC warm-up, as presented in Figure 3. This factor is associated with species that are only partially converted by the TWC, and with by-products of incomplete oxidation.

### 3.2. NMVOC EFs

Figure 4 presents the NMVOC EF distributions for the three vehicles classified by carbon number and chemical families during the Artemis cold urban and motorway cycles. Compounds ranging from C_1_ to C_22_ were lumped into 15 chemical families: unsaturated aliphatics, linear alkanes, branched alkanes, cycloalkanes, bicycloalkanes, monoaromatics, naphthenic monoaromatics, dihydronaphthalenes, naphthalenes, alcohols, carbonyls, unsaturated carbonyls, acids, other oxygenated compounds, and nitrogen compounds. EFs of chemical families for each vehicle are presented in Table 2. Detailed EF lists for each family, as shown in Figure 4, are presented in Appendix A for GDI CU and MW cycles; Appendix A for PFI CU and MW cycles; and Appendix A for diesel CU and MW cycles. 

#### 3.2.1. Gasoline vehicle EFs

The GDI passenger car is the highest emitting vehicle with a total NMVOC EF of 252 ± 65 and 15 ± 8 mg/km for the CU and MW cycles, respectively. Gasoline emissions mostly occur during the cold start and decrease by a factor of 17 during the MW cycle. Previous studies on Euro 5 GDI generally show large discrepancies with NMHCs ranging from 22 to 221 mg/km for Euro 5 GDI vehicles during NEDC cycles at 23 and −7 °C, respectively [67], while a recent work reports 250 mg/km [57], in good agreement with our data. The GDI vehicle tested here seems to be one of the highest NMVOC emitters of its category. The PFI vehicle is the second highest emitting vehicle, with a total NMVOC EF of 23.5 mg/km and 0.4 mg/km for the CU and MW cycles, respectively. These EFs are 10 to 50 times lower than the EFs observed for the GDI vehicle. Emissions from the PFI vehicle are relatively low compared to previous studies on Euro 5 PFI vehicles [76]. 

For both gasoline vehicles, monoaromatic compounds represent 60 and 68% of the total NMVOC EFs during the CU cycle, respectively, and 51 and 69% during the MW cycle. These results are in agreement with previous studies, where similar NMVOC speciation is found for a variety of PFI and GDI vehicles [24]. The high aromatic content reported here can be partly explained by the lack of data for small alkanes < C_5_ and alkenes < C_2_, which can represent up to 50% of the NMVOC emissions for gasoline vehicles [24]. As shown in Figure 4a–d, the two gasoline vehicles exhibit different monoaromatic distributions. During CU cycles, the GDI emission of monoaromatics is dominated by C_9_ compounds, mainly composed in descending order of propylbenzene, 1,2,4-trimethylbenzene, and 1-ethyl-3-methylbenzene. They are followed by C_8_ compounds such as m+p-xylenes, o-xylene, and ethylbenzene; followed by toluene and benzene; and, finally, C_10_ to C_11_ monoaromatics. Similar monoaromatic distribution is emitted during the MW cycles, apart from benzene, which outruns toluene, and 1,2,4-trimethylbenzene, the highest emitted C_9_ monoaromatic. For the PFI vehicle, the monoaromatic emissions are dominated by C_8_ compounds in the order m+p-xylene > o-xylene > ethylbenzene. They are followed by C_9_ compounds mainly composed in descending order of 1,2,4-trimethylbenzene, 1-ethyl-4-methylbenzene, and 1-ethyl-2-methylbenzene; followed by toluene and benzene; and, finally, C_10_ to C_16_ compounds. Aromatics from C_11_ to C_16_ are only detected by the PTR-ToF-MS and for the PFI vehicle. These results suggest that the introduction of GDI technology will not lower monoaromatic emissions, but rather increase their concentration in urban environments. 

Alkanes are the second most emitted class of pollutants for both PFI and GDI vehicles, representing 15 and 28% of the total NMVOCs during the CU cycle, respectively, and 18 and 42% during the MW cycle, respectively. Alkanes are lower for the PFI than for the GDI vehicle, even though branched alkanes were measured for the GDI vehicle only, for which they represent 84 and 87% of the total alkanes for CU and MW cycles, respectively. During the CU cycle, the alkane distribution for the PFI vehicle is dominated by hexane, followed by a general decreasing pattern up to docosane. For the GDI vehicle, the alkane distribution is characterized by a peak around C_9_, mainly represented by C_6_ to C_13_ branched alkanes, while linear alkanes distribution spans from C_6_ to C_22_ and presents a maximum at C_8_ (octane), followed by a tail distribution up to C_22_. Total oxygenated compounds emitted by the PFI vehicle represent 10 and 7% of the NMVOC EFs during the CU and MW cycles, respectively. Contributions from oxygenated compounds are lower for the GDI vehicle, representing only 3% of the NMVOC EFs during both the CU and MW cycles. Carbonyl compounds are the most abundant species for both gasoline vehicles. For the PFI vehicle during the CU cycle, emissions are dominated by acetaldehyde followed by benzaldehyde, acetone, acetophenone, acrolein, methacrolein, and formaldehyde. Similar compounds are observed for the GDI vehicle, as presented in Appendix A. Alcohols such as ethanol, methanol, and phenol also represent an important fraction of the oxygenated compounds, essentially during the CU cycle. Ethanol is a component of the fuel and can be up to 10% in volume (SP95-E10 fuel). Its presence in the emission is therefore linked to unburnt fuel. Here, ethanol EF accounts for 1 and 1.5% of the total NMVOC EFs for the GDI and the PFI vehicle, respectively. Finally, PFI emissions comprise traces of carboxylic acids from C_2_ to C_5_, as well as benzoic acid.

#### 3.2.2. Diesel Vehicle EFs

NMVOC EFs for the diesel vehicle are 15.3 ± 6.7 and 2.1 ± 0.5 mg/km during the CU and MW cycles, respectively, making it the least emitting of the three tested vehicles. Most of the emissions occur during the cold start. This work confirms previous observations for Diesel Euro 5 vehicles, which reported EFs varying from 13.5 to 44.9 mg/km during cold start NEDC cycles [67], 13.6 to 70.3 mg/km during cold NEDC cycles [77], and 20 ± 10 mg/km during Artemis CU cycles [55]. Our results for the MW cycle are also in agreement with previous works reporting 2.0 and 2.3 mg/km for two Euro 5 diesel vehicles [57].

Contrary to gasoline vehicles, emissions from the diesel vehicle are dominated by oxygenated compounds which may reach 67 and 68% of the NMVOC EFs during CU and MW cycles, respectively. These compounds are represented by saturated carbonyls (36–60%), carboxylic acids (28 and 21%), unsaturated carbonyls (4%), and alcohols (1–3%). During the CU cycle, acetaldehyde, acetone, and formaldehyde are the most abundant compounds followed by C_4_ to C_7_ carbonyls. Acetic acid accounts for 2.7 and 0.5 mg/km during the CU and MW cycles, respectively. Other acids such as formic acid, C_3_ acids, and C_4_ acids are only detected during the CU cycle. Unsaturated carbonyls from C_3_ to C_6_ and C_3_ to C_8_ are measured during the CU and MW cycles, respectively. Alcohols are mainly represented by methanol and ethanol during the CU cycle only, and methanol was measured in the diesel fuel headspace (Appendix A). A variety of other oxygenated trace compounds are also detected in the diesel vehicle emissions, such as glyoxal, furan, pyran, and anhydrides (including maleic and phthalic anhydride). As previously shown, the majority of these oxygenated compounds are emitted during the CU cycle after the catalyst activation (Figure 2a, factor 4) or during the MW cycle (Figure 2b, factor 4), and seem to be formed on the DOC. 

Aliphatic compounds are the second most emitted species, representing 23 and 28% of the total NMVOC EFs during the CU and MW cycles, respectively. During the CU cycle, linear alkanes (49%) and unsaturated aliphatics (40%) are the most abundant, followed by cycloalkanes (8%) and bicycloalkanes (3%). The alkane distribution for the CU cycle is comprised between C_6_ and C_16_, with a maximum at C_10_ with decane (372 µg/km). Cycloalkanes distribution spans from C_6_ to C_13_ with a maximum at C_8_ (81 µg/km), while bicyloalkanes range from C_8_ to C_15_ with a maximum around C_11_ (21 µg/km). During the MW, ratios of linear alkanes and bicycloalkanes increase to 79 and 4% of the total aliphatic EF, respectively. On the other hand, ratios of unsaturated aliphatics, cycloalkanes, and bicycloalkanes decrease to 15 and 3%, respectively.

#### 3.2.3. IVOC Characterization

In light of Drozd et al.’s methodology [17], IVOCs are classified into three categories: aliphatic IVOCs comprising linear, branched, and unsaturated aliphatics; single-ring aromatic IVOCs comprising alkyl-substituted monoaromatics and bicyclic compounds with only one aromatic ring; and general IVOCs comprising all the compounds that do not fall in the two first categories.

The IVOC fraction for the gasoline vehicles spans from 2.7% for the GDI UC to 13% for the PFI MW cycles. GDI and PFI IVOC composition is similar and mostly comprises aliphatic and general IVOCs. Aliphatic IVOCs, and, more specifically, linear alkanes, are the main contributors to gasoline vehicle IVOCs, representing from 46 to 80% of IVOC emissions. General IVOC emissions comprise essentially naphthalene and vary from 14% of total IVOCs for the GDI UC to 42% for the GDI MW cycles. These results are in line with those of previous studies on IVOCs from gasoline vehicles [17,19]. Based on the PMF analysis results, the IVOC fraction is mainly associated with GDI UC’s factor 6 (Figure 2c) and is expected to become preponderant a few minutes after the start of the vehicles, as described in Figure 3.

Diesel IVOC fractions range from 5.8% during the CU cycle to 9.5% during the MW cycle. It is essentially composed of aliphatic IVOCs. Linear alkanes represent 95 and 92% of the diesel IVOC fraction for the CU and MW cycles, respectively. These results are in agreement with previous studies on diesel vehicles [20]. The IVOC fraction is generally higher during the MW cycle, due to the lower removal efficiency of the DOC catalyst toward IVOCs compared to other NMVOCs with higher volatility. This behavior was already observed for gasoline vehicles [19].

### 3.3. Comparison with COPERT Emission Inventories

COPERT (EMISIA SA [78]) is one of the standard computational tools used to calculate road transport emissions in the EU, as detailed in *The EMEP/EEA Air Pollutant Emission Inventory Guidebook* [79]. It is commonly used to estimate emission inventories [80]. COPERT provides vehicle fleet and activity road transport data, particularly emission factors for all major pollutants, heavy metals, particulate matter, and greenhouse gases for various vehicle categories. Concerning NMVOCs, COPERT provides a complete speciation for gasoline and diesel vehicles. However, this speciation is old and was determined for Euro 1 vehicles. For that reason, COPERT cannot differentiate GDI from PFI cars, but only provides one general speciation for all gasoline vehicles. COPERT speciation accounts for 60 compounds, against 147, 116, and 85 detected in this study for the GDI, PFI, and diesel vehicles, respectively. Appendix A illustrate the complete NMVOC speciation from COPERT emission inventory for gasoline passenger cars and diesel cars, respectively. COPERT speciation does not detail alkanes > C_12_ and aromatics > C_13_, which are considered as relevant SOA precursors. Thus, 31% of COPERT diesel emissions are unspeciated compounds > C_12_. The speciation provided here is much more detailed, with compounds up to C_22_, and differentiates the emissions from GDI and PFI vehicles. Moreover, this work provides updated EFs for some compounds of interest, such as naphthalene and phenol, which are not well defined or listed in COPERT database. Given the rapid changes in engine and aftertreatment systems technologies, detailed speciation for each Euro standards, including IVOCs, is essential to improve the urban air quality models. This speciation should be used to upgrade European road transport emission inventories, such as COPERT.

## 4. Conclusions

This work is the first to present PMF analysis of highly time-resolved PTR-ToF-MS measurements (1s resolution) of vehicle emissions. Three Euro 5 vehicles (one diesel, one GDI, and one PFI vehicle) were tested on a roll-bench chassis dynamometer on Artemis driving cycles. PMF analysis was used to investigate the influence of the engine status and aftertreatment devices on the exhaust VOC emissions. During the first few hundred seconds of the CU cycle, NMVOC emissions were mainly associated with unburnt fuel. The effect of the aftertreatment devices became evident during and after catalyst activation, which led to an overall decrease in exhaust pollutants. Concomitant emissions of incomplete oxidation products, produced by both the DOC and the TWC, as well as aliphatic and monoaromatic compounds in the C_10_–C_16_ range, probably desorbed from the engine manifold and aftertreatment systems, were observed during the warm-up of the catalysts. The diesel vehicle presented high and non-repeatable emissions of acetic acid occurring indifferently during the CU and MW cycles. Concerning the GDI vehicle, emissions of benzene, aliphatic fragments, and CO were observed simultaneously with strong accelerations during the CU cycle. These emissions were associated with the low performances of the TWC toward NMVOCs during fuel-rich combustion conditions.

This study provides an updated inventory of NMVOC EFs calculated by merging datasets from online PTR-ToF-MS measurements and offline tenax cartridges (GC-MS) analysis. More than a hundred compounds have been identified and quantified for each vehicle, including oxygen- and nitrogen-containing VOCs, and aliphatic and aromatic compounds from C_6_ to C_22_ with their isomers. The current work indicates that the GDI vehicle can emit up to 10 and 16 times more than the PFI and diesel vehicles, respectively. The diesel vehicle mostly emits products of incomplete combustion, unburnt fuel components, and heavy aliphatic and aromatic compounds. 67% of its emissions are oxygenated compounds, more particularly toxic carbonyl compounds such as formaldehyde and acetaldehyde. Gasoline emissions were dominated by C_9_ and C_8_ monoaromatics for the GDI and the PFI vehicles, respectively, and alkanes ranging from C_6_ to C_22_. Most of the gasoline emissions derived from unburnt fuel emitted during the first minutes of the CU cycle. These results clearly suggest that the introduction of the GDI technology may potentially increase monoaromatics concentration in urban environments.

The IVOC fraction spans from 2.7% of the NMVOCs for the GDI, 9.5% for the diesel, and 13% for the PFI vehicle. Diesel IVOCs are essentially linear alkanes associated with unburnt fuel emissions. Concerning gasoline vehicles, the IVOC fraction is composed of linear alkanes and naphthalene, representing up to 42% of the total IVOC EFs. The IVOC fraction becomes preponderant after the start of the vehicles, as the removal efficiency of the catalysts seems to be better for volatile NMVOCs than for heavier IVOCs.

Together with aromatic EFs, emission reports of IVOCs are a key issue to assess the SOA formation potential of Euro 5 vehicles and their impact on urban air quality. The exhaustive NMVOC speciation provided by this work represents an upgrade with respect to existing inventories in the COPERT database, which is based on older vehicles and does not differentiate PFI vehicles from GDI vehicles. Moreover, COPERT gives no information concerning the speciation of aliphatic and aromatic compounds > C_13_. Thus, this work should be considered for current emission inventories.

Future research on vehicular emissions should broaden the variety of studied vehicles and combine different analytical techniques. In doing so, chemistry models will have access to exhaustive NMVOC inventories, including oxygen- and nitrogen-containing VOCs and IVOCs. Moreover, PMF analysis of online VOCs data should be applied to more varied driving conditions, such as idle, creep, or real-world driving. These investigations could provide useful information on specific emissions associated with the functioning of vehicles, their origin, and hints about where to act specifically for their reduction. 

## Figures and Tables

**Figure 1 toxics-10-00184-f001:**
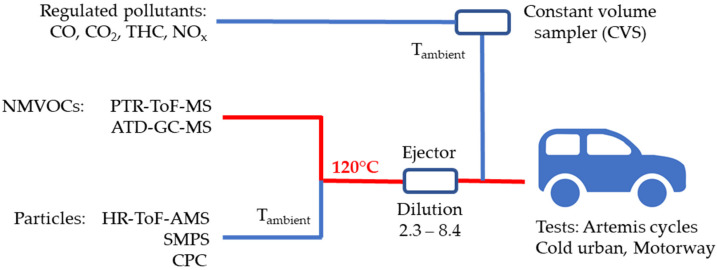
Schematic of the experimental setup.

**Figure 2 toxics-10-00184-f002:**
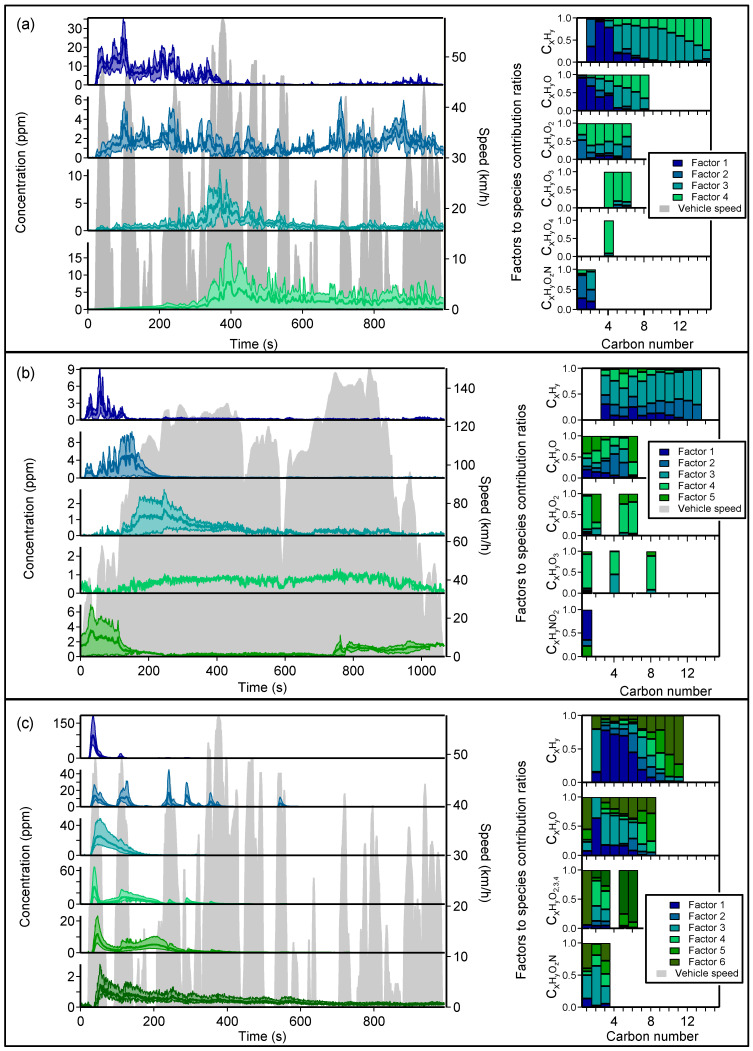
PMF factor temporal variations and contribution ratios for a typical (**a**) diesel CU cycle, (**b**) diesel MW cycle, and (**c**) GDI CU cycle. Time series of the factors are averaged over (**a**) 4 cycles, (**b**) 2 cycles, and (**c**) 6 cycles. For each factor, the bold line represents the averaged concentration corrected from the dilution, and the colored zone represents the associated standard deviation. The gray zone represents the speed variations during CU and MW cycles. Factors to species contribution ratios are classified by carbon, oxygen, and nitrogen number.

**Figure 3 toxics-10-00184-f003:**
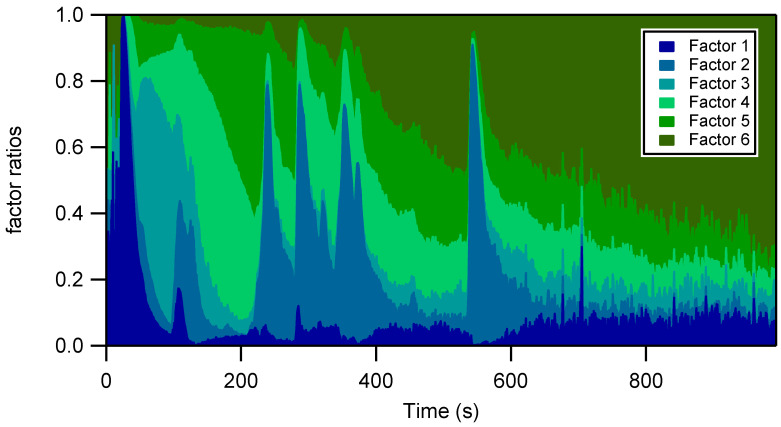
Temporal evolution of the PMF factor concentration ratios (ppm/ppm) during the GDI cold urban cycle.

**Figure 4 toxics-10-00184-f004:**
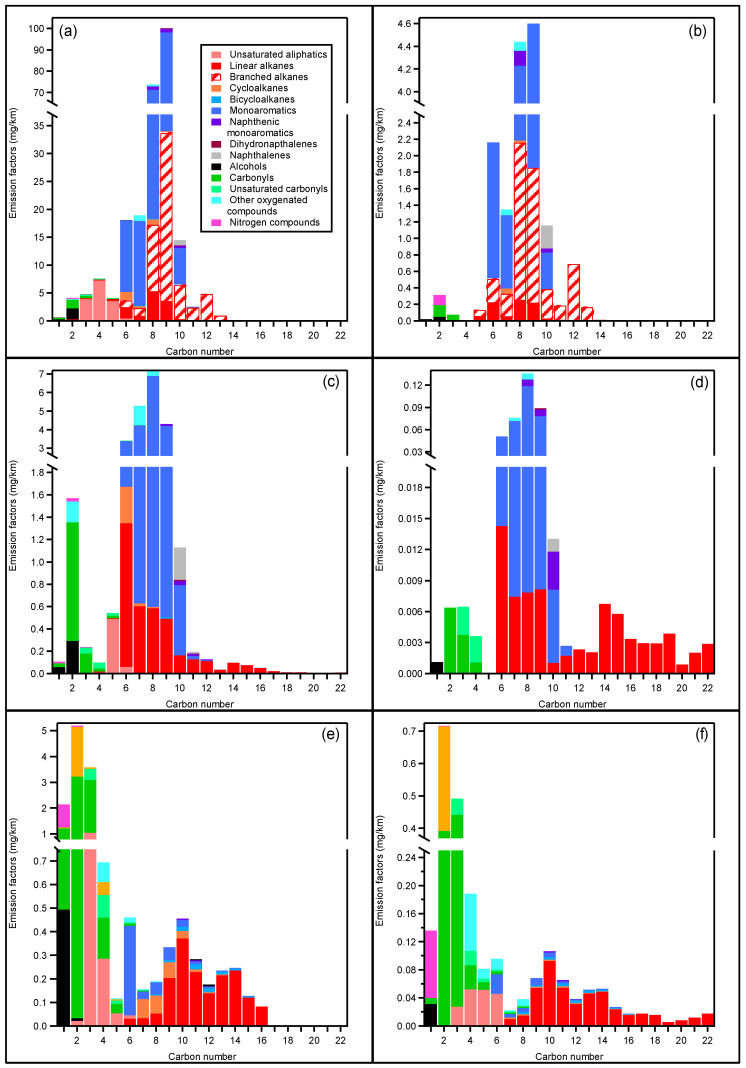
NMVOC EFs in mg/km as a function of carbon numbers for a typical (**a**) GDI CU cycle, (**b**) GDI MW cycle, (**c**) PFI CU cycle, (**d**) PFI MW cycle, (**e**) diesel CU cycle, and (**f**) diesel MW cycle. The different colors correspond to the main chemical classes measured by PTR-ToF-MS and GC-MS.

**Table 1 toxics-10-00184-t001:** Technical characteristics of the tested vehicles and experimental conditions.

		Diesel Euro 5	Gasoline Euro 5	
Vehicle Name		D	PFI	GDI
Size class		2.0 HDI	1.0 VVTI	1.2 TCE 16 V
Engine capacity (cm^3^)		1997	998	1149
Weight (kg)		1515	1030	1100
Odometer mileage (km)		103000	27712	97089
Catalyst type		DOC	TWC	TWC
Particulate filter type		FBC-DPF	-	-
GC-MS dilution ratio		2.3	2.3	8.4
PTR-ToF-MS dilution ratio		7.8–15	7.8–8.4	18.5–23.4
Tests ambient temperature (°C)		25 ± 2	23 ± 2	20 ± 2
Road loads	a_0_ (N)	124.78	88.68	98.1
	a_1_ (N/(m/s))	0	0	0
	a_2_ (N/(m/s)^2^)	0.515	0.381	0.429

**Table 2 toxics-10-00184-t002:** Synthetic overview of the total NMVOC EFs and detailed chemical family EFs for the three Euro 5 vehicles during cold urban and motorway cycles. * EFs annotated with an asterisk are calculated from PTR-ToF-MS data, while other EFs are calculated from ATD-GC-MS data.

	Emission Factors					
	Gasoline PFI + TWC		Gasoline DI + TWC		Diesel DOC + FBC DPF	
Compound Class	Cold Start (mg/km)	Motorway (µg/km)	Cold Start (mg/km)	Motorway (mg/km)	Cold Start (mg/km)	Motorway (µg/km)
Total NMVOC	23.6	420 ± 25	251 ± 64	15.4 ± 8.2	14.9 ± 6.7	2230 ± 1050
Total aromatics	16.5	315 ± 9	155 ± 34	8.3 ± 4.8	0.6 ± 0.2	80 ± 45
Monoaromatics	16.0	290	150 ± 32	7.8 ± 4.6	0.6 ± 0.2 *	70 ± 40
Dihydronaphthalenes	0.01 *	2 ± 1 *	0.5 ± 0.2 *	0 *	0 *	0 *
Naphthenics monoaromatics	0.17 *	22 ± 7 *	3.7 ± 1.2 *	0.18 ± 0.08 *	0.012 ± 0.007 *	4 ± 2 *
Naphthalenes	0.30	1 ± 1 *	0.9 ± 0.7 *	0.3 ± 0.1 *	0 *	6 ± 3
Total aliphatics	4.6	76	88.2 ± 27.1	6.5 ± 3.1	3.4 ± 1.3	600 ± 290
Unsaturated aliphatics	0.56	0	15 ± 6	0	1.4 ± 0.4 *	90 ± 40 *
Alkanes	3.6	76	70 ± 20	6.4 ± 3.0	1.7 ± 0.8 *	470 ± 230
Cycloalkanes	0.42	0	3 ± 1	0.1 ± 0.1	0.28 ± 0.09 *	15 ± 5 *
Bicycloalkanes	0.04 *	0 *	0.2 ± 0.1 *	0 *	0.01 ± 0.04 *	25 ± 16 *
Total oxygenated	2.4	29 ± 16	7.7 ± 3.1	0.5 ± 0.3	10 ± 5	1450 ± 695
Alcohols	0.36 *	1 ± 2 *	2.4 ± 1.2 *	0.07 ± 0.05 *	0.5 ± 0.1 *	31 ± 15 *
Carbonyls	1.4 *	11 ± 8 *	2.6 ± 0.9 *	0.2 ± 0.1 *	6.2 ± 1.6 *	770 ± 370 *
Unsaturated carbonyls	0.15 *	5 ± 2 *	0.6 ± 0.2 *	0 *	0.5 ± 0.2 *	80 ± 20 *
Acids	0.03 *	0 *	0 *	0 *	2.9 ± 2.9 *	460 ± 260 *
Others	0.49 *	12 ± 4 *	2.1 ± 0.8 *	0.2 ± 0.1 *	0.1 ± 0.1 *	110 ± 30 *
Total nitrogen	0.065	0	0.27 ± 0.12	0.06 ± 0.03	0.9 ± 0.2	98 ± 16
Nitroalkanes	0.02 *	0 *	0.07 ± 0.02 *	0 *	0.9 ± 0.2 *	98 ± 16 *
Nitriles	0.04 *	0 *	0.2 ± 0.1 *	0.06 ± 0.03 *	0.03 ± 0.01 *	0 *

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
