# Peer review of "Detailed Speciation of Non-Methane Volatile Organic Compounds in Exhaust Emissions from Diesel and Gasoline Euro 5 Vehicles Using Online and Offline Measurements"

_toxics, 2022, doi:10.3390/toxics10040184_

Round 1
Reviewer 1 Report
Dear Authors,
My general comment concerns the content of the manuscript, in which the authors refer very often to the material in the supplement, which makes analysis of the manuscript content much more difficult. It would be beneficial to consider supplementing selected material contained in the supplement to the manuscript.
I propose also to take into account detailed remarks:
- It would be beneficial to supplement the manuscript with a list of abbreviations.
- Abstract: It should emphasize the novelty of the manuscript.
- Introduction: It would be beneficial to explain why the Authors chose to study emissions in Artemis cycles?
- Materials and methods: In the description of the tests it would be beneficial to present the test stand illustrating the methodology of the tests carried out. Furthermore, please provide information on ambient temperatures at which the tests were performed on a chassis dynamometer and the road load of the dynamometer.
- Materials and methods: Line 169: Is the flow rate value of 400 cm^3/min correct given a sampling line of 1 mm inner diameter?
- Materials and methods: It would be helpful if you could complete the description of Equation 2 - Tcycle. Also, please provide the units of the quantities appearing in the equation.
- Results and discussion: Does the description of "cold urban cycle speed", which refers to changes in speed as a function of time, require the addition of the term "cold" in the legend description? After all, the driving cycle is independent of the engine temperature at the beginning of the test.
- Results and discussion: Figure 1 b: Please correct the description of the values on the vertical axis (values overlap).
- Results and discussion: 3.2. The subsection should not begin with Figure 3.2, but should be preceded by text. Figure 2 shows the NMVOC average run emission values. I suggest replacing the term "Emission factors" with "Average emissions".
- Results and discussion. Lines 446-448: The authors unnecessarily repeat the reference to Figure 2.
- Results and discussion. Lines 452-453: Please verify that surely the lower NMHCs emission values were obtained for a temperature of -7 degrees Celsius?
- Page 16: Please complete the descriptions of Figures S7 and S8 in the "Supplementary materials" list.
Best regards
Reviewer 2 Report
In my opinion this article is interesting and worthy. The article deals with topical environmental problems of gasoline and diesel engines. The authors propose an original method for monitoring and evaluating the environmental performance of engines. The experimental procedure is described in sufficient detail. The graphic material of the article is designed with high quality and competently. The article presents original data on the environmental performance of gasoline and diesel engines. The authors carried out a detailed analysis of the obtained data. Conclusions are drawn and recommendations are given.
However, the article has a few shortcomings. To accept this paper for publication in the toxics, some improvements and revisions are required as specified in the review.
Article notes:
1. I think that the title of the article should not contain abbreviations.
2. Using a list of lumped references is not very helpful for a reader (for example, ... [1-4] on page 1 or [26-29] on page 2). Assessment/justification should be provided for each reference, even it may be short.
3. There is no information about the measuring instruments (and accuracy), and there is also no data on the error (uncertainty) of the experiment. The authors provide only a link to their previous work. I believe that brief information about the instrumentation base should be present in this article at least.
4. It is necessary to bring restrictions on the proposed methodology for assessing the environmental performance of engines (engine size, engine type, purpose, fuel type, operating conditions, etc.). 5. The manuscript need more clear and formal, first person like “I, We” should be avoided.
6. Please formulate the directions for further research on this topic in the "Conclusions" section.
The present form of the article needs to be revised and supplemented as it has been suggested above.
Reviewer 3 Report
Dear Authors,
The presented research demonstrates a high effort and a good quality. The literature review, the presentation of the measurements, and the evaluation methods are detailed and precise.
However, the article in this form is hard to read and process. There should be more subsections. Many topics should be divided to separate sections.
For instance, tables or figures would help see through the measurement system. There is plenty of information in the paragraphs, but it is difficult to search for them if somebody wants to repeat or check the work.
Also, the results should be re-classified. The depicted figures are informative, and they are demonstrative without the article. However, the connection between the text and the figures should be more spectacular.
With these changes, the article can reach more readings.
Round 2
Reviewer 1 Report
Dear Authors,
In order to improve the quality of the article, I suggest including the following points:
- In the description to equation (2) the Authors gave the distance for the urban part (UC) 4.51 km and for the motorway part (MW) 23.8 km. Why these values do not correspond to those specified for the Artemis cycle, which, according to numerous sources (e.g. https://dieselnet.com/standards/cycles/artemis.php) are: for the urban part 4.874 km; for the motorway part 28.737 (Motorway 130) and 29.547 (Motorway 150)? Such a big difference for the freeway section has a significant influence on emission factors determined from equation (2).
- Table 2: How was the average EF of Total NMVOC determined? Are the mean values of EF Total NMVOC the sum of the mean values of the individual groups viz: EF Total Aromatics, EF Total aliphatics, EF Total oxygenated and EF Total nitrogen? If yes, please explain the differences of these values (EF Total NMVOC and sum for each group) for Motorway (Diesel).
- For better clarity of the article content for the reader, it would be beneficial to include a list of abbreviations at the end of the article rather than in the supplementary material. At the same time, the list of designations should be presented in alphabetical order.
- Incorrect literature references "Error! Reference source not found." (lines 134, 548, 549).
- The description of tables and figures should be standardized in the text of the manuscript - written with the first capital letter.
- In the text of the manuscript, the Authors enter website addresses (e.g., line 192, line 589). These should be replaced with bibliographic references.
Best regards
